# To Be, or Not to Be: That Is the Hamletic Question of Cryptic Evolution in the Eastern Atlantic and Mediterranean *Raja miraletus* Species Complex

**DOI:** 10.3390/ani13132139

**Published:** 2023-06-28

**Authors:** Alice Ferrari, Valentina Crobe, Rita Cannas, Rob W. Leslie, Fabrizio Serena, Marco Stagioni, Filipe O. Costa, Daniel Golani, Farid Hemida, Diana Zaera-Perez, Letizia Sion, Pierluigi Carbonara, Fabio Fiorentino, Fausto Tinti, Alessia Cariani

**Affiliations:** 1Department of Biological, Geological and Environmental Sciences, University of Bologna, 40126 Bologna, Italy; alice.ferrari6@unibo.it (A.F.); valentina.crobe2@unibo.it (V.C.); alessia.cariani@unibo.it (A.C.); 2Department of Life and Environmental Sciences, University of Cagliari, 09126 Cagliari, Italy; rcannas@unica.it; 3Branch Fisheries Management, Department Agriculture, Forestry and Fisheries, Cape Town 8018, South Africa; roblesliesa@hotmail.com; 4Institute for Biological Resources and Marine Biotechnology, National Research Council, 91026 Trapani, Italy; fabrizio.serena@irbim.cnr.it (F.S.); fabio.fiorentino@irbim.cnr.it (F.F.); 5Laboratory of Marine Biology and Fisheries, Department Biological, Geological and Environmental Sciences, University of Bologna, 61032 Fano, Italy; marco.stagioni3@unibo.it; 6Centre of Molecular and Environmental Biology (CBMA) and ARNET-Aquatic Research Network, Department of Biology, University of Minho, Campus de Gualtar, 4710-057 Braga, Portugal; fcosta@bio.uminho.pt; 7Department of Evolution, Systematics and Ecology, The Hebrew University of Jerusalem, Jerusalem 9190401, Israel; dani.golani@mail.huji.ac.il; 8Ecole Nationale Supérieure des Sciences de la Mer et de l’Aménagement du Littoral, Campus Universitaire de Dely Ibrahim, Algiers 16320, Algeria; hemidafarid@yahoo.fr; 9Institute of Marine Research, 5817 Bergen, Norway; diana.zaera-perez@hi.no; 10Department of Biosciences, Biotechnologies and Environment, University of Bari Aldo Moro, 70125 Bari, Italy; letizia.sion@uniba.it; 11COISPA Technology and Research, 70126 Bari, Italy; carbonara@coispa.it; 12Stazione Zoologica Anton Dohrn, 90149 Palermo, Italy

**Keywords:** cartilaginous fish, brown skate, conservation biology, population genetics, mtDNA, microsatellite loci

## Abstract

**Simple Summary:**

The *Raja miraletus* species complex exhibits high levels of morphological and ecological stasis along with the antipodean distribution in the Eastern Atlantic and Indian Oceans. We investigated genetic variability and differentiation between taxa and geographical populations by integrating mitochondrial and nuclear DNA markers. The extraordinary occurrence of at least five different sibling taxa in the Northeastern Atlantic Ocean and Mediterranean Sea is documented, supporting cryptic speciation and stabilising selection.

**Abstract:**

Despite a high species diversity, skates (Rajiformes) exhibit remarkably conservative morphology and ecology. Limited trait variations occur within and between species, and cryptic species have been reported among sister and non-sister taxa, suggesting that species complexes may be subject to stabilising selection. Three sibling species are currently recognised in the *Raja miraletus* complex: (i) *R. miraletus* occurring along the Portuguese and Mediterranean coasts, (ii) *R. parva* in the Central-Eastern Atlantic off West Africa and (iii) *R. ocellifera* in the Western Indian Ocean off South Africa. In the present study, the genetic variation at mitochondrial and nuclear markers was estimated in the species complex by analysing 323 individuals sampled across most of its geographical distribution area to test the hypothesis that restricted gene flow and genetic divergence within species reflect known climate and bio-oceanographic discontinuities. Our results support previous morphological studies and confirm the known taxonomic boundaries of the three recognised species. In addition, we identified multiple weakly differentiated clades in the Northeastern Atlantic Ocean and Mediterranean, at least two additional cryptic taxa off Senegal and Angola, a pronounced differentiation of ancient South African clades. The hidden genetic structure presented here may represent a valuable support to species’ conservation action plans.

## 1. Introduction

The evolutionary debate on the nature of species boundaries [1,2] is based on paradigms such as Mayr’s discontinuous variation and reproductive isolation of species and Darwin’s continuity between varieties, geographical populations and species [2]. Natural hybrid zones and secondary contacts with gene introgression unequivocally show that species boundaries have a semipermeable nature [3] and that intrinsic barriers to gene flow (i.e., pre- and postzygotic barriers) are in some cases incomplete. Species life cycles, ecological features and adaptive phenotypes are particular key points influencing species distribution and dispersal within the marine environment [4,5], in which permanent and intermittent breaks (e.g., landmasses, unsuitable habitats, upwelling areas and oceanographic fronts) may isolate populations, enabling ecological differentiation [6,7], genetic divergence [8], reproductive isolation and speciation [9,10,11]. Molecular systematics and phylogenetics have greatly contributed to the assessment of relationships among taxa and effectively contributed to delineate the hierarchy of evolutionary frames in which recently diverged taxa exhibit, on average, lower divergence than taxa in the later stages of the speciation process [6]. Over the last two decades, increased evidence has emerged for speciation governed by entirely different mechanisms, leading to so-called sibling or cryptic species (sensu Bickford [12]). The idea that species can evolve into similar morphologies is well established [13], but the use of molecular delimitation methods has now brought cryptic species to the forefront in many research arenas [6,14,15]. Bickford et al. [12] identified at least two recurrent themes in animals wherein morphological distinctiveness and reproductive isolation are unpaired: in groups using non-visual mate-recognition signals (e.g., chemical, olfactory, acoustic, electric-field senses) and in groups living under environmental conditions that promote the stabilising selection of morphological traits (e.g., extreme habitats, specialised host–parasite relationships, deep-sea environments, fishery pressure). Among elasmobranchs, skates and rays exhibit highly effective modulation of electro-sensory signals depending on behaviour ([16] and references therein). At the same time, they display a marked conservation of ecological and morphological traits [17,18,19,20,21,22,23], especially between recently diverged species [12]; a strong evolutionary success in terms of resilience at the evolutionary scale [24] and a high degree of endemism [25,26] and species richness [27,28,29].

Investigations into the role of biogeographical barriers on the speciation of marine organisms have increasingly concentrated across several taxa [30,31,32]. Prior to 2016, the brown skate *Raja miraletus* Linnaeus, 1758 was thought to be distributed throughout the Mediterranean Sea and from northern Portugal, along the western and south-eastern coasts of Africa [33,34]. This distributional range is much wider than expected for a small-sized rajid, given the limited potential for dispersal in a species with a relatively sedentary behaviour of adults and juveniles [35,36,37,38,39] and the lack of egg dispersal [40]. Nominal *R. miraletus* exhibits a pronounced benthic ecology, with most records from 10 m to 150 m on sandy and hard bottoms [25,33] and a generalist feeding behaviour [41,42]. Due to its high and stable abundance over its distribution, small body size and early maturation (age at maturity estimated at 2.7 years; [43]), it was considered highly resilient to exploitation and was assessed globally as Least Concern in the International Union for Conservation of Nature Red List [44,45]. Since then, *Raja ocellifera* Regan 1906 has been resurrected for the southern African population [28,46] and was assessed as Endangered in 2020 [46]. The newly described *R. parva* Last and Séret 2016 from west Africa has not been assessed [28]. 

The *R. miraletus* complex exhibits a high level of stasis of the general external morphology over its range; all populations exhibit a distinctive bright tricolored (blue, black and yellow) eyespot on the upper ochre-brownish surface at the base of each pectoral fin [34,47]. After the first identification of three parapatric or allopatric groups (Mediterranean, West Africa and South Africa) based on the variation of morphometric and meristic characters [48], preliminary evidence of cryptic speciation in *R. miraletus* was observed by integrating results from mitochondrial DNA analysis, morphology and host–parasite relationships from specimens collected in Central-Southern Africa [49,50]. *Raja miraletus* was then recognised as a species complex of at least four valid taxa based on combined data for *COI* and *NADH2* [51]: (1) the northernmost *R. miraletus*, occurring in the Mediterranean Sea and adjacent North-Eastern Atlantic waters, (2) the southernmost and resurrected *R. ocellifera* (associated with mtDNA data from Naylor et al. [49]; *Raja* cf. *miraletus 1*, NCBI Accession Number JQ518895, [49]) occurring, in the Western Indian Ocean, off South Africa, and in the Indian Ocean, from False Bay to Durban, (3) the central African *R. parva*, distributed from Senegal to Angola (associated with mtDNA data from Naylor et al. [49]; *Raja* cf. *miraletus 2*, NCBI Accession Number JQ518890, [49]) and (4) a still undescribed taxon (*Raja miraletus*, NCBI Accession Number JQ518891, [49]), occurring from Mauritania to Senegal where it is therefore sympatric with *R. parva*.

The advent of high-throughput DNA sequencing technologies and the launch of global DNA-based biodiversity assessments (e.g., DNA barcoding; [52]) has provided raw data, enabling the determination of taxonomic, ecological and evolutionary aspects of cryptic and sibling species, where the term “sibling” denotes a cryptic species with a recent common ancestor, implying a sister species relationship [53] and even more challenging conservation issues [12]. Moreover, molecular methods coupling markers obtained from mitochondrial DNA (mtDNA) and nuclear DNA (nuDNA) have improved the resolution of species boundaries and revealed gene introgression/hybridisation phenomena in marine fish including elasmobranchs [10,54,55,56]. Understanding the liaison between species life-history traits, ecology and the adaptive phenotypes leading to hidden population divergence and reproductive isolation is of utmost importance for skates, whose conservation is often hampered by the lack of species-specific data [57].

This study uses the integrative support of the mtDNA *cytochrome oxidase subunit I* barcode region (*COI*) and eight nuDNA *EST-linked* polymorphic microsatellite loci (*EST-SSRs*; [58]) to estimate the genetic variation among 323 specimens collected across almost the full geographic range of the *R. miraletus* species complex [40,47] and exhibiting the typical tricoloured eyespots. We tested hypotheses of the relationship between restricted gene flow and genetic divergence within the species complex, specifically in relation to climatic and oceanographic discontinuities. Additionally, we sought to establish parallel patterns between our findings and variations in morphology and parasite prevalence, which were independently assessed [48,50]. As compared to previous knowledge, our findings contributed to describe a richer scenario concerning the taxonomic units and zoogeographic boundaries characterising the *R. miraletus* complex.

## 2. Materials and Methods

### 2.1. Sampling

A total of 323 specimens of from the *R. miraletus* species complex were collected between 2000 and 2014 (Table 1 and Appendix A) during scientific research programs (South Africa, Angola and Mediterranean Sea) by contracted commercial fishermen (Senegal, Levantine Sea and Israel) or at local fish markets (Algeria). Scientific trawl surveys were carried out in South Africa (2006 and 2011, Africana cruises), Angola (2006, Nansen cruises), the whole Mediterranean Sea (2000–2014 Mediterranean International Trawl scientific Surveys, MedITS; [59]) and national scientific trawl surveys (2000–2010 Italian Gruppo Nazionale Demersali surveys, GruND; [60]; the 2007 Portuguese scientific surveys of the Instituto Português de Investigação do Mar) allowed for a comprehensive sampling, covering most of the wide geographical distribution of the *R. miraletus* complex (Figure 1). All individuals were easily assigned to the *R. miraletus* complex based on their very distinctive morphotype and species-specific diagnostic characters [25,40]. Fin clips and muscle tissues were cut from each individual using sterile tweezers and clippers, transferred to a clean tube filled with 96% ethanol and stored at −20 °C for subsequent DNA analyses.

### 2.2. Genetic Data Analysis

Detailed protocols used for DNA extraction, PCR amplification, DNA sequencing and genotyping of mitochondrial and nuclear markers [56,58,61,62] are described in the Appendix A.

#### 2.2.1. Genetic Diversity

A total of 281 *COI* newly generated sequence electropherograms were manually edited and aligned by CLUSTAL W software [63] implemented in MEGA v.11 [64]. The presence of stop codons and sequencing error was verified through amino acidic translation [65]. Individual *COI* sequences were first compared with sequences deposited in public repositories in order to confirm their phylogenetic identity and rule out any error due to mishandling of samples on board or during the laboratory activities, namely GenBank (http://www.ncbi.nlm.nih.gov/genbank/, accessed on 21 May 2023) through the BLAST algorithm (http://blast.ncbi.nlm.nih.gov/Blast.cgi, accessed on 21 May 2023) and the Barcode of Life Data System (BOLD), using the BOLD identification engine ([66]; http://www.boldsystems.org, accessed on 21 May 2023). A total of 41 additional homologous *COI* sequences of *R. miraletus* complex were retrieved from both databases selecting records from different geographical locations (South Africa, Namibia, Strait of Sicily, Aegean Sea and Israel) when metadata giving the collection area were accessible ([26,67,68,69,70,71,72,73,74,75,76,77,78,79]; Table 1; see Appendix A for more detail). The retrieved sequences were aligned with those newly generated and a final dataset of 322 homologous *COI* sequences was obtained.

The number of polymorphic sites (S), the number of haplotypes (H), the haplotype diversity (Hd), the nucleotide diversity (π; [80]) and their standard deviations were calculated using DNASP v.6 [81]. The haplotype frequencies were estimated using ARLEQUIN v.3.5.2.2. [82].

The average genetic distances observed within and between the two identified Central–Southern African and the Northeastern Atlantic–Mediterranean clades of *R. miraletus* complex were calculated using the Tamura-Nei (1993) model implemented in MEGA as the best evolutionary substitution model following the corrected Akaike Information Criterion (AICc; [83]). Genetic distances were then compared with the range of those estimated among other congeneric species. For this, we retrieved homologous *COI* sequences of the following species public databases (NCBI and BOLD): *Raja straeleni* Poll 1951, *Raja microocellata* Montagu 1818, *Raja asterias* Delaroche 1809, *Raja brachyura* Lafont 1873, *Raja clavata* Linnaeus 1758, *Raja montagui* Fowler 1910, *Raja polystigma* Regan 1923, *Raja radula* Delaroche 1809 and *Raja undulata* Lacepède 1802 ([26,84]; Appendix A).

A total of 256 chromatograms for each of the eight *EST-SSR* loci were obtained and manually inspected using GENEMAPPER v.5 (Applied Biosystems, Waltham, MA, USA). Allele calling and binning were performed with GENEMAPPER. The presence of null alleles, stuttering and allele drop-out was tested using MICRO-CHECKER v.2.2.3 [85] with 1000 randomisations on Bonferroni correction. The multilocus *EST-SSR* genotypes were analysed using GENETIX v.4.05 [86] to estimate observed (H_O_) and expected heterozygosity (H_E_) and the number of alleles (A). Jack-knifing over loci was performed to assess the single-locus effects on Weir and Cockerham’s F-statistics estimators. The deviation from the Hardy–Weinberg equilibrium (HWE) and Linkage Disequilibrium (LD) was investigated using GENEPOP on the web v.4.2 [87]. Allelic richness (Ar) and inbreeding coefficient (F_IS_) were estimated using FSTAT v.2.9.3.2 [88].

#### 2.2.2. Population Connectivity and Phylogenetic Signals

The phylogenetic relationships among individual haplotypes were inferred by the TCS method implemented in the software POPART [89]. The graphical representation of the resulting network has been modified with Adobe Photoshop.

Population connectivity within the *R. miraletus* species complex was investigated by estimates of Φst and Fst values using ARLEQUIN with 10,000 permutations, *p* < 0.05. The Tamura-Nei (1993) substitution model was applied to the mtDNA dataset to estimate Φst values. Genetic heterogeneity among the geographical samples was also assessed by a hierarchical analysis of molecular variance (AMOVA, [90]). Significance was assessed using a null distribution of the test statistic generated by 10,000 random permutations of the individuals in the samples. The significance threshold of the pairwise comparisons (*p* < 0.05) was adjusted with the sequential Bonferroni correction for multiple simultaneous comparisons [91] implemented in the R package “sgof” [92].

To unravel the individual-based genetic clustering, the *COI* and the *EST-SSR* datasets were analysed using the Bayesian algorithm implemented in BAPS v.6.0 [93] and STRUCTURE v.2.3.4 [94], respectively. The latter analysis on SSRs loci was carried out assuming an admixture ancestry model with the geographical origin of samples as prior information (LOCPRIOR models), associated with a correlated allele frequencies model. For each simulation of K (1–20), five independent replicates were run, setting a burn-in of 200,000 iterations and 500,000 iterations for the Markov Chain Monte Carlo (MCMC) simulation. Cluster matching and permutation were performed using CLUMPAK [95], while the most likely value for K was estimated from the mean log probability of the data using four alternative statistics (medmedk, medmeak, maxmedk and maxmeak) carried out using STRUCTURESELECTOR [96].

Discriminant analysis of principal components (DAPC) using the R package Adegenet v.2.0.1 [97] was implemented in R v.4.0.5 (R Core Team [98]) using sampling locations as a priori groups (K = 5). Then, the optimal number of PCs to use in the DAPC was determined using the optim.a.score() command.

Phylogenetic relationships between and within the Central–Southern African and Northeastern Atlantic–Mediterranean *COI* lineages were estimated using a Bayesian coalescent approach, implemented in BEAST v.1.10.4 [99]. Sequences of *R. undulata* (BOLD record ELAME177-09, NCBI accession numbers KT307412, KT307413, KT307414), the closest related species to the *R. miraletus* complex, were used as outgroups. The Bayesian reconstruction was obtained using the Hasegawa, Kishino and Yano (HKY + G) model of evolution [100], as the most appropriated model inferred by MEGA software, a strict molecular clock model, the Yule Process as species tree prior, the Piecewise linear and constant root as population size prior. To ensure convergence of the posterior distributions, an MCMC run of 60,000,000 generations sampled every 1000 generations with the first 25% of the sampled points removed as burn-in was performed. We analysed the log file using TRACER v.1.7.2 [101] to calculate the robustness of the posterior distributions for all parameters and recover average divergence time and 95% confidence intervals. The plausible trees obtained with BEAST were summarised using the program TREEANNOTATOR and the resulting phylogenetic relationships among population samples and the posterior probabilities at nodes were visualised with FigTree v.1.4.4 (available at http://tree.bio.ed.ac.uk/software/figtree/, accessed on 21 May 2023) and edited with the iTol v.6.7.5 online tool [102]. 

Cryptic species were also delimited by using two different methods: the distance-based method “Automatic Barcode Gap Discovery” (ABGD; [103]) computed on the online web application (http://wwwabi.snv.jussieu.fr/public/abgd/abgdweb.html, accessed on 23 May 2023), using default values, and the phylogenetic-based method Bayesian Poisson Tree Process (bPTP, [104]), conducted on the web server (available at http://species.h-its.org/ptp/, accessed on 23 May 2023) with 100,000 MCMC generations, a thinning interval of 100% and 10% of burn-in.

## 3. Results

### 3.1. Genetic Diversity

The *COI* dataset was a total of 322 sequences, while the *EST-SSR* dataset was made up of a total of 256 individuals overall distributed in 14 geographic samples (Table 1 and Appendix A). The final *COI* alignment consisted of 529 nucleotide positions and included 76 variable sites (14.3%) and 64 parsimony informative sites (12.1%). On average, *COI* polymorphism showed low estimates of nucleotide diversity (π) and very high haplotype diversity (Hd), with ANG being the most polymorphic sample (Hd = 0.858 ± 0.041 SD, π = 0.02543 ± 0.00380 SD, K = 13.453; Appendix A). Thirty-nine haplotypes were found and none were shared between samples from the Northeastern Atlantic–Mediterranean and Central–Southern African Regions (Figure 2 and Appendix A). The average Tamura-Nei genetic distances (DTN) among geographical samples of the Northeastern Atlantic–Mediterranean were extremely low (DTN = 0.0025 ± 0.0011 SE; Table 2), while those observed among geographical samples of the Central–Southern Africa were an order of magnitude higher (mean DTN = 0.0183 ± 0.0029 SE). The DTN between Northeastern Atlantic–Mediterranean and Central–Southern Africa samples were much higher (mean DTN = 0.0733 ± 0.0117 SE) and is comparable between species distances recorded among species in the genus *Raja* (Table 2).

Summary statistics of the eight polymorphic microsatellite loci per geographical sample and over all the loci considered are shown in Appendix A. Mean allelic richness (Ar_mean_) ranged from 1.242 (POR) to 1.724 (SEN). After Bonferroni correction, significant LD was not detected between any pair of loci, and the average mean observed and expected heterozygosity (H_O_/H_E_) for the eight loci was 0.259/0.392. After applying the Bonferroni correction, significant HWE departures were found over all loci in several geographical samples, apart from SEN, POR, SIC and ISR. The Portuguese sample was monomorphic at five loci (LERI 26, LERI 34, LERI 63, LERI 40 and LERI 44). Overall, MICRO-CHECKER results detected the presence of scoring errors such as stuttering and null alleles in all loci (Appendix A), regardless of the geographical sample. Nevertheless, we did not exclude any of them since Jack-knife analysis did not reveal outliers outside of the confidence interval (Appendix A).

### 3.2. Population Connectivity and Phylogenetic Signals

Caution should be applied when interpreting the results obtained here due to the small sample size for some localities and the subsequent decrease in the discriminatory power of the analyses.

The TCS network of the *COI* haplotypes (Figure 2) identified two main haplogroups, differentiated by at least 30 mutations and corresponding to the Central–Southern African and the Northeastern Atlantic–Mediterranean samples. The former haplogroup included 23 haplotypes that grouped into four largely differentiated geographic clusters located off Senegal, Angola/Namibia/South Africa and two off Angola. The Senegalese cluster formed only by the SEN sample (N = 5) showed three slightly differentiated private haplotypes. In contrast, the Angolan sample (ANG, N = 27) showed strongly differentiated haplotypes grouped into two endemic Angolan subclusters together and a third cluster shared with the South African (SAF, N = 40) and Namibian (NAM = 3) samples. The Northeastern Atlantic–Mediterranean haplogroup included 16 weakly divergent haplotypes (Figure 2 and Appendix A). Four of them were shared by several samples and areas: (i) the haplotype Hap_24 was shared by Portuguese, Algerian and Strait of Sicily samples; (ii) the most frequent Hap_25 was shared by samples from Algeria, Balearic Islands, Sardinia, Strait of Sicily, Tuscany and Adriatic Sea; (iii) the Hap_28 was shared by samples from Algeria, Strait of Sicily and Adriatic Sea; iv) the Hap_32 was shared by Adriatic and Greek samples. In contrast, the Eastern Mediterranean samples from the Israeli coast (Hap_37 and Hap_38) and Levantine Sea (Hap_39) yielded only three endemic haplotypes.

Most of the pairwise Φst values among 14 geographical samples based on *COI* data were significant even after the Bonferroni correction was applied (Appendix A). High levels of differentiation were observed between the African and Northeastern Atlantic–Mediterranean samples, as well as between the Western and Eastern Mediterranean. The *EST-SSR* data showed a similar pattern of genetic differentiation in terms of Fst calculated over 12 macro areas, even after the Bonferroni correction was applied (Appendix A).

The hierarchical AMOVA found the highest percentage of molecular variation among groups with four sample groupings (AMOVA 3: Southern Africa vs. Angola vs. Senegal vs. Northeastern Atlantic–Mediterranean Sea; Appendix A) when using the *COI* dataset, and with five sample groups (40.06%, AMOVA 4: Southern Africa vs. Angola vs. Senegal vs. Portugal–West Mediterranean vs. Eastern Mediterranean) when using the AMOVA 5 (six groups) explained the genetic variation among samples with the proportion of the genetic variation among populations within very low groups (2.58% with mitochondrial data and 7.72% with the *EST-SSR* data).

The Bayesian clustering analysis based on mtDNA data (Figure 3a) revealed six genetic clusters: the first cluster (green) included individuals from SAF, NAM and ANG; the second (purple) and third clusters (light blue) ANG; the fourth cluster (red) was unique to SEN; the fifth (yellow) and sixth (blue) clusters were unique to the Northeastern Atlantic and Mediterranean with individuals from POR, BAL, SAR and TUS in the fifth cluster and those from GRE, ISR and LEV in the sixth cluster. Individuals from ALG, SIC and ADR were randomly associated with the fifth and the sixth clusters, suggesting the southern Mediterranean, especially the Siculo-Tunisian area, as a potential admixture zone of the latter clusters.

The outputs of the STRUCTURE analysis based on *EST-SSR* data analysed with STRUCTURESELECTOR did not provide clear-cut evidence of the most likely number of clusters using four alternative statistics (medmedk, medmeak, maxmedk and maxmeak), while the maximum value of ΔK was verified with K = 3 (Appendix A). Thus, the results from K = 2 to K = 7 were assessed with CLUMPAK (Appendix A). The barplot of the clustering K = 2 supported the separation of the Central–Southern Africa samples from Northeastern Atlantic–Mediterranean Sea and with an admixed genetic composition of the Senegalese individuals (Appendix A). The clustering K = 3 further discriminated between the Angolan sample from those of South Africa, as well as samples from the Western and Eastern Mediterranean Sea (Appendix A). At the same time, the Angolan sample displayed an intermediate genetic composition between the South African and Senegalese clusters. This trend was resolved by the clustering K = 4 (Figure 3b and Appendix A), corresponding to the best grouping revealed by AMOVA (Appendix A). The plot showed a deep differentiation of the Northeastern Atlantic–Mediterranean samples east to the Strait of Sicily.

The DAPC computed on *EST-SSR* data with sampling locations used as a priori group identified 10 optimal numbers of PCs and separated five main clusters: South African, Angolan, Senegalese, Northeastern Atlantic–Western Mediterranean and Eastern Mediterranean clusters, with the South African and Angolan clusters partially overlapping (Appendix A).

Bayesian approach using MCMC simulation was used to test for a speciation signal (Yule process) between lineages from Central–Southern Africa and Northeastern Atlantic–Mediterranean (Figure 4, see Appendix A for haplotype distribution among samples). All effective sample size (ESS) values exceeded 200, indicating a solid evaluation of all parameters. The model based on the substitution rate estimated for mtDNA showed a clear separation between haplotypes from Central–Southern Africa (Hap_1 to Hap_23) and the Northeastern Atlantic–Mediterranean (Hap_24 to Hap_39; Figure 4). The phylogenetic relationships among lineages and haplotypes were congruent with the relationships obtained with the parsimony network results (Figure 2). Furthermore, within the main Central–Southern African lineage, four clusters of haplotypes were reconstructed with high posterior probability (PP = 1): the most basal Angolan haplotype H_20; a second cluster formed by six Angolan haplotypes (Hap_14 to Hap_19); a Senegalese cluster (Hap_21 to Hap_23); and a South African/Angolan/Namibian cluster formed by all the South African haplotypes (Hap_1–9), the Namibian haplotype Hap_10, the Angolan haplotypes (Hap_12 and Hap_13) and the Angolan/Namibian haplotype (Hap_11). 

The two species delimitation approaches (the ABGD and bPTP methods) yielded the same result (Figure 4). Five groups have been identified: four are formed by samples from the Central–Southern Atlantic, in agreement with the phylogenetic reconstruction, and one formed by samples from the Northeastern Atlantic and Mediterranean.

## 4. Discussion

The *R. miraletus* species complex is widely distributed, occurring from the Mediterranean Sea down the west coast of Africa to South Africa. The geographically isolated population off the south coast of South Africa was originally described as *R. ocellifera* Regan, 1906, but was synonymised with *R. miraletus* [105]. An extensive morphological study by McEachran et al. [48] found a marked differentiation between specimens from South Africa and those from Mediterranean elements, whilst the West African samples, in particular those from Angola, showed intermediate meristic features. However, they considered the differences between the Mediterranean and South African populations to be clinal and concluded that *R. miraletus* is a polymorphic species with three partially separated populations. Strikingly stable gross morphology has always been misleading for taxonomists. Subsequent integrated molecular and morphometric studies have shown that the three partially separated populations are valid species [51]. *Raja ocellifera* has been resurrected [44] and Last and Séret [51] described a new species, *R. parva* from Liberia and Angola. Last and Séret [51] stated that *R. parva* differed from the Senegalese *Raja* cf. *miraletus* 1 (sensu Naylor et al. [49]) in total body length, a shorter snout and a smaller tail, and they suggest that two further putative species (or taxa) occur off Senegal, Guinea, Liberia, down to Angola. Among the West African samples, the *R. parva* were the most distinct, even though some characters were intermediate between the Mediterranean and South African specimens [48].

The recent designation of the *R. miraletus* species complex [51] increased interest in an evolutionary and phylogenetic investigation of the complex based on more extensive sampling and analysis of combined nuclear and mitochondrial genetic data. Ferrari et al. [73] inferred population structure within *R. miraletus* (sensu stricto) across the Mediterranean Sea based on an analysis of nucleotide variation in three mtDNA markers, while Crobe et al. [67] preliminarily recognised four divergent *COI* lineages from the Eastern Atlantic populations of this morphologically conserved taxon. It should be noted that our study (i) was based on an unprecedently high number of specimens unequivocally assigned to the *R. miraletus* species complex; (ii) included specimens collected from the localities within the distribution ranges of the four putative taxa occur and (iii) was based on an integrated analytical approach that combines sequence variation of both mtDNA (the universal *COI* barcode region) and nuDNA(allelic variation in eight polymorphic *EST-linked* microsatellite loci) markers. This methodology enables our study to advance the molecular characterisation of this Hamletic taxon and to increase the knowledge on its status in the Mediterranean Sea.

The mitochondrial and microsatellite data consistently agreed in genetically defining the taxonomic and geographical boundaries of *R. miraletus* (sensu stricto), which are distributed in the entire Mediterranean Sea and the adjacent Northeastern Atlantic Ocean, at least in the Portuguese coastal waters. The estimated mean Tamura-Nei genetic distance between the Northeastern Atlantic–Mediterranean clade and the Central–Southern African clade (DTN = 0.0735) was in the upper range of corresponding pairwise interspecific estimates among several congeneric *Raja* species supporting a specific level of differentiation.

Surprisingly, the host–parasite specificity established between different African population of the *R. miraletus* species complex, with species from the genus *Echinobothrium* (Cestoda: Diphyillidea) highlighted by Caira et al. [50], supports the clades identified by Naylor et al. [49] based on molecular data. Specimens of *R. ocellifera* (*R.* cf. *miraletus* 1 from South Africa from [49]) hosted *E. yiae* Caira, Rodriguez and Pickering, 2013, those of *R. miraletus* from Senegal hosted *E. mercedesae* Caira, Rodriguez and Pickering, 2013 and specimens of a second clade off Senegal likely belonging to *R. parva* (*R.* cf. *miraletus* 2 sensu Naylor et al. [49]) hosted two additional new species [50]. These findings were partially supported by our study, where the Senegalese sample (SEN) is a distinct subunit (Figure 2 and Figure 3; Hap_21, Hap_22 and Hap_23 in Figure 4; Φst in Appendix A). Unfortunately, the very limited sampling along the Senegalese coast (N = 5) prevents any definitive conclusions.

Overall, our results emphasised the strong differentiation between South Atlantic-Indian *R. ocellifera* and the Northeastern Atlantic–Mediterranean *R. miraletus* and justifies the resurrection of the former taxon. The complex oceanographic conditions along the African coast with alternating cold and warm currents, from north to south cold Canary Current, warm Angola Current, cold Benguela Current and warm Agulhas Current, undoubtedly played a role in speciation along the African coast. Likewise, the complexity oceanographic and geological discontinuities characterising the Eastern Atlantic and the Mediterranean Basin may likely influence phylogeography, population structure and connectivity as well as evolution at multiple taxonomic levels [48,106]. Oceanographic heterogeneities, such as current systems, play a key role for ecologically divergent natural selection in elasmobranchs, such as the ecological radiation of the genus *Pseudobatos* Last, Séret and Naylor, 2016 in the Gulf of California, strongly influenced by habitat heterogeneity and the geological history of the region [107]. This condition seemed to be true not only for skates, but even more so for African coastal bony fish (i.e., genus *Argynosomus* De la Plyaie, 1835), whose evolutionary histories, including the dispersal phase, have been influenced by the Benguela Current [108,109]. The Benguela Current region ranges from Cape Agulhas to Cape Frio, where the north-flowing cold Benguela Current meets the south-flowing warm Angola current (see Hirschfeld et al. [106], Figure 2d,f) for current model map details). The cold waters of the Benguela system are likely to have strongly reduced gene flow between *R. ocellifera* and *Raja* cf. *miraletus* (sensu Naylor et al. [49]). This diversification was evident from the level of mean sequence divergence observed between the two geographical populations estimated at 7.3%, a value comparable to, or higher than, any divergence measured between *R. undulata* and other congeneric taxa. On the other hand, the intertropical Canary current inflowing from the northeast could have influenced the diversification of the Senegalese taxa, whose migration southwards would be hampered by the intermittent Cape Blanc upwelling area. This upper boundary could have influenced and limited gene flow in Northern Mauritania. Like other skates [10,110,111] and teleost species [112], no genetic differentiation was observed between Northeastern Atlantic and Mediterranean populations of *R. miraletus*. This suggests that the Strait of Gibraltar did not represent an effective barrier to gene flow, rather than an accession gate to ancient *refugia*. In contrast, Melis et al. [113] found moderate significant population differentiation between the Mediterranean and the Atlantic Ocean in the congeneric thornback skate *R. clavata*, suggesting an effective role of the Strait in limiting the dispersal of individuals. 

The slight genetic population structure observed within the Mediterranean Sea represents a true novelty for this species complex. The unforeseen regional East–West Mediterranean structure highlighted by nuclear markers (Figure 3b and Appendix A) could be linked to bathymetry and hydrogeological fronts or discontinuities. In particular, the shallow bathymetry characterising the Southwest part of the Mediterranean, coupled with the species’ preferences for continental shelf habitats, may likely enhance the dispersal of brown skate. The area ranging from the easternmost part of Sicily and the adjacent geo-morphological depression of the Calabrian Arc (down to 3000 m) is dominated by cyclonic/anticyclonic inversions of water masses. The combination of these features could have limited gene flow between western and eastern populations and driven the differentiation of the Eastern Mediterranean samples (ADR, ISR and LEV; Figure 1). The specific habitat and depth preferences, the less pronounced migratory behaviour and the limited dispersal capability of taxa belonging to the *R. miraletus* species complex are rather common characteristics among skates [62,73,113,114], although other congeneric species did not show such evidence of deep differentiation at both nuDNA and mtDNA markers [10,56,72,110]. In detail, the Mediterranean starry ray *R. asterias* showed a strongly structured population with three geographical clades corresponding to the western, central–western and central–eastern Mediterranean areas [62]. *R. clavata* showed a weak but detectable phylogeographic structure in the Levantine Sea [73] and a finer structuring located off the Algerian coasts and Tyrrhenian basins, suggesting the occurrence of additional barriers to dispersal [113]. On the contrary, *R. polystigma* showed a slightly differentiated Adriatic haplotype but a near panmictic population in the central–western part of the basin [56]. These different patterns of population structure in such closely related species can be explained by their bathymetric range which drive different ecological features [72]. 

## 5. Conclusions

This study coupled a massive and extensive sampling effort covering the full distribution of the *R. miraletus* species complex with analysis of genetic variability in both mtDNA and nuDNA markers, individual clustering, phylogeography and variance at different population levels. The results were partially congruent with previous taxonomic and meristic analyses. The use of both nuclear and mitochondrial markers resulted in identifying signals of species differentiation and in supporting the existence of at least five cryptic taxa within the *R. miraletus* species complex, four of which have been previously suggested with scattered genetic data. In addition to the evolutionary meaning of this evidence, genetics is shown to aid conservation efforts by revealing hidden diversity that deserves special attention and in the monitoring of taxa that are important fishing bycatch species. The new insights highlighted in the present research paper suggest that the extraordinary intraspecific diversity observed across such a wide geographical scale should be carefully considered to update or set dedicated and effective measures to reduce the impact of skate bycatch during fishing activities and improve their conservation.

## Figures and Tables

**Figure 1 animals-13-02139-f001:**
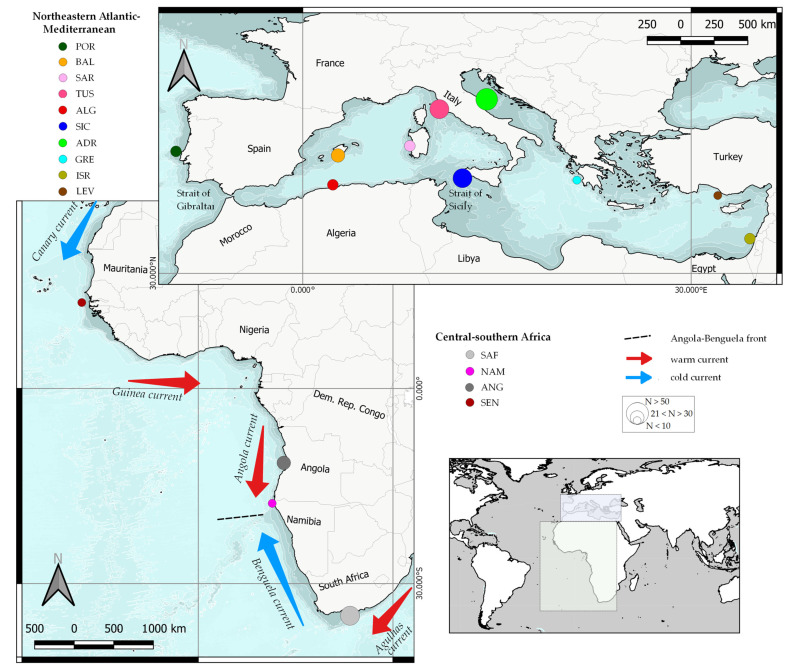
Sampling sites of *Raja miraletus* species complex collected in the Northeastern Atlantic–Mediterranean and the Central–Southern Africa regions. A simplified representation of the main oceanic currents of the Eastern Atlantic is indicated. Acronyms of the Macro Area Codes are given as in Table 1. Colours in agreement with the haplotype network legend of Figure 2. Different dimensions of circles are related to sample size.

**Figure 2 animals-13-02139-f002:**
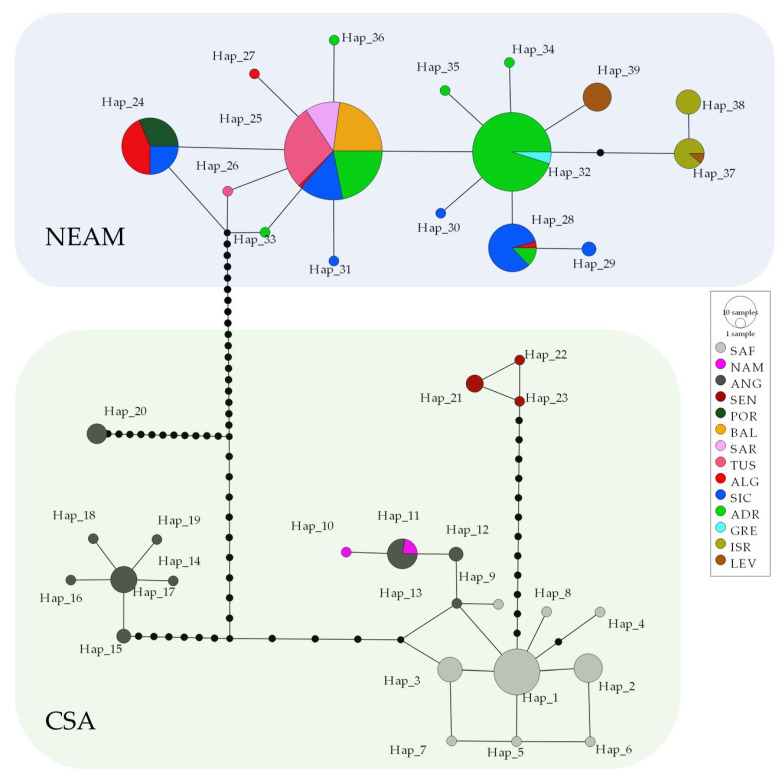
TCS network of *cytochrome oxidase subunit I (COI)* haplotypes shown by *Raja miraletus* across most of its distribution area. CSA, Central–Southern Africa; NEAM, Northeastern Atlantic–Mediterranean Sea. Circles are proportional to haplotype frequencies. Black dots between branch nodes indicate substitutions. Black circles at network nodes represent unsampled haplotypes. Acronyms of the Macro Area Codes are given in Table 1.

**Figure 3 animals-13-02139-f003:**
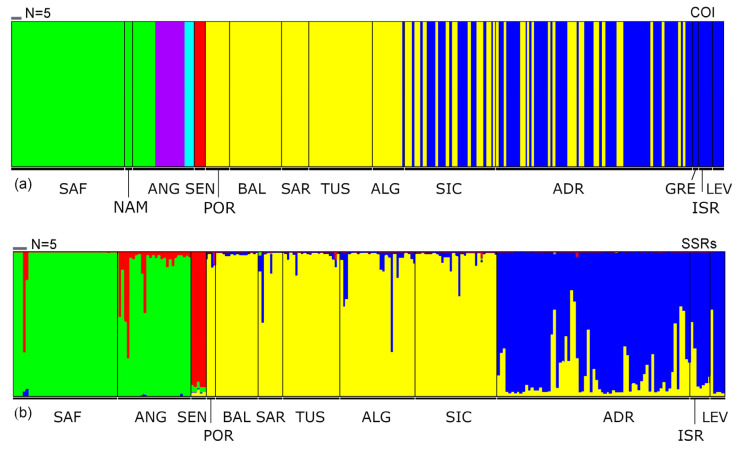
Bayesian admixture analysis among individuals belonging to the *Raja miraletus* species complex across most of its geographical distribution area. (**a**) Distribution of the *cytochrome oxidase subunit I* clades in the population samples inferred with BAPS; each colour represents one distinct haplogroup (cluster), and each bar represents a different individual. (**b**) Results of the Bayesian individual clustering using STRUCTURE results for K = 4; each vertical bar represents one individual, in which a different colour represents the estimated cluster membership. Acronyms of the Macro Area Codes are given in Table 1.

**Figure 4 animals-13-02139-f004:**
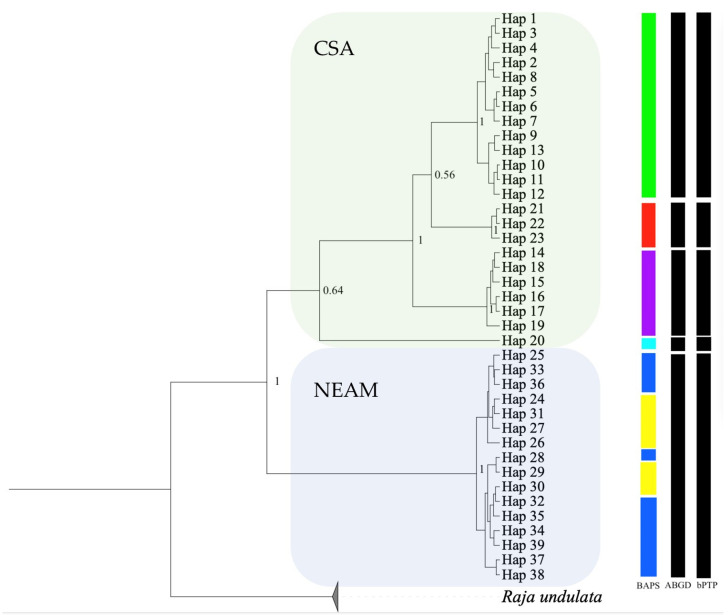
Bayesian coalescent tree summarising phylogenetic relationships between the Central–Southern African (CSA) and Northeastern Atlantic–Mediterranean (NEAM) *COI* lineages. *Raja undulata* was used as an outgroup (BOLD record ELAME177-09, NCBI accession numbers KT307412, KT307413, KT307414). Posterior probability (≥0.5) values are reported on nodes. Coloured bars near haplotype nodes refer to the genetic clusters identified by BAPS results (Figure 3a). The outcomes of species delimitation analyses using ABGD and bPTP methods are shown as vertical bars on the right.

**Table 1 animals-13-02139-t001:** Sampling data and locations. “N (tot)”, refers to the total number of individuals sampled in this study, according to the methods indicated in the Source column. Of these “N (*COI*)” and “N (*EST-SSRs*)” refer to individuals *COI*-sequenced and genotyped in this study. N (tot) = 0 when available *COI* sequences were retrieved from public databases and integrated into the mtDNA dataset, as specified in the “Source” column. The last row refers to geographical samples previously described in McEachran et al. 1989 [48]. 1—Mediterranean group. 2—Mauritania and Senegal group. 3—Gulf of Guinea–equatorial African group. 4—Angolan sample. 5—South African sample.

Sampling Area	Macro Area Code	N (tot)	N (*COI*)	N (*EST-SSRs*)	Years	Source (Trawl Survey Program)	McEachran et al., 1989 [48]
Central-Southern Africa (CSA)							
South Africa—South Coast	SAF	8	6	8	2006	ST (Africana)	5
South Africa—South Coast		0	5	0	2007, 2012	GB/BOLD	5
South Africa—South Coast		32	30	31	2011	ST (Africana)	5
Namibia	NAM	0	3	0	2009, 2010	GB/BOLD	5
Angola	ANG	27	27	26	2006	ST (Nansen)	4
Senegal	SEN	5	5	5	2007	CF	2
Northeastern Atlantic–Mediterranean Sea (NEAM)							
Portugal	POR	3	3	3	2007	ST (IPIMAR)	n.a.
Portugal		0	7	0	2005, 2007	GB/BOLD	n.a.
Balearic Islands	BAL	19	19	16	2006	ST (MedITS)	1
Balearic Islands		0	3	0	2013	ST (MedITS)	1
Sardinia	SAR	11	11	8	2002, 2005	ST (MedITS)	1
Tuscany	TUS	26	22	21	2005, 2006	ST (MedITS)	1
Tuscany		16	6	13	2008, 2010	ST (MedITS)	1
Algeria	ALG	8	8	5	2002, 2003	FM (Algiers)	1
Algeria		10	9	8	2009, 2010	FM (Algiers)	1
Strait of Sicily—Adventura Bank	SIC	22	22	22	2014	ST (MedITS)	1
Strait of Sicily—Maltese Bank		16	12	8	2000, 2002	ST (MedITS)	1
Strait of Sicily—Maltese Bank		0	11	0	2007	GB/BOLD	1
Adriatic Sea—Northern Italian coast	ADR	39	31	20	2006, 2007	ST (MedITS; GruND)	1
Adriatic Sea—Croatian coast		24	24	8	2002, 2004	ST (MedITS; GruND)	1
Adriatic Sea—Southern Italian coast		19	16	19	2004	ST (MedITS; GruND)	1
Adriatic Sea—Albanian coast		19	13	17	2004	ST (MedITS; GruND)	1
Ionian Sea		4	3	4	2004	ST (MedITS; GruND)	1
Greece—Aegean coast	GRE	0	3	0	2014	GB/BOLD	1
Israel	ISR	8	7	7	2009	CF	1
Israel		0	3	0	2012	BOLD	1
Israel		0	4	0	2014	GB/BOLD	1
Levantine Sea	LEV	0	2	0	2016	GB/BOLD	1
Levantine Sea		7	7	7	2009	CF	1

n.a.: not available. ST: Scientific Trawl survey. CF: Contracted Fishermen. FM: Fishery Market. GB: GenBank Database. BOLD: Barcoding of Life Database.

**Table 2 animals-13-02139-t002:** Mean genetic distances within (in grey, in diagonal) and between congeneric *Raja* species. Standard error values for distances between species are reported above the diagonal. CSA, Central–Southern Africa; NEAM, Northeastern Atlantic–Mediterranean Sea. The mean genetic distance between NEAM and CSA is indicated in bold.

	*Raja asterias*	*Raja brachyura*	*Raja clavata*	*Raja microocellata*	*Raja montagui*	*Raja polystigma*	*Raja radula*	*Raja straeleni*	*Raja undulata*	*Raja miraletus* (CSA)	*Raja miraletus* (NEAM)
*Raja asterias*	0.0025 ± 0.0012	0.0139	0.0106	0.0132	0.0135	0.0121	0.0095	0.0111	0.0121	0.0155	0.0189
*Raja brachyura*	0.0863	0.0030 ± 0.0016	0.0107	0.0094	0.0119	0.0114	0.0121	0.0125	0.0136	0.0141	0.0172
*Raja clavata*	0.0591	0.0514	0.0038 ± 0.0000	0.0115	0.0103	0.0113	0.0070	0.0059	0.0141	0.0133	0.0169
*Raja microocellata*	0.0877	0.0465	0.0598	0.0000 ± 0.0000	0.0125	0.0107	0.0114	0.0115	0.0127	0.0140	0.0172
*Raja montagui*	0.0829	0.0606	0.0511	0.0638	0.0000 ± 0.0000	0.0066	0.0126	0.0116	0.0128	0.0132	0.0172
*Raja polystigma*	0.0748	0.0526	0.0514	0.0596	0.0229	0.0000 ± 0.0000	0.0108	0.0112	0.0123	0.0119	0.0164
*Raja radula*	0.0487	0.0685	0.0280	0.0648	0.0646	0.0564	0.0010 ± 0.0012	0.0071	0.0119	0.0137	0.0173
*Raja straeleni*	0.0593	0.0593	0.0150	0.0601	0.0590	0.0558	0.0282	0.0019 ± 0.0010	0.0122	0.0131	0.0174
*Raja undulata*	0.0767	0.0736	0.0733	0.0799	0.0790	0.0711	0.0742	0.0735	0.0009 ± 0.0010	0.0122	0.0170
*Raja miraletus* (CSA)	0.1053	0.0857	0.0861	0.0947	0.0830	0.0778	0.0908	0.0907	0.0770	0.0183 ± 0.0029	0.0117
*Raja miraletus* (NEAM)	0.1072	0.1020	0.0989	0.1007	0.0921	0.0897	0.0974	0.1033	0.0959	**0.0733**	0.0025 ± 0.0010

## Data Availability

The newly produced sequence data that support the findings of this study are openly available in GenBank at https://www.ncbi.nlm.nih.gov/genbank (accessed on 27 June 2023) reference number OR193802–OR194004. The *EST-SSR* data that support the findings of this study are available as a genotype matrix from the corresponding author upon request.

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
