# Peer review of "To Be, or Not to Be: That Is the Hamletic Question of Cryptic Evolution in the Eastern Atlantic and Mediterranean Raja miraletus Species Complex"

_animals, 2023, doi:10.3390/ani13132139_

Round 1
Reviewer 1 Report
The manuscript “To be, or not to be: that is the Hamletic question of the extraordinary cryptic evolution in the Raja miraletus species complex (Chondrichthyes: Elasmobranchii)” is a well written and interesting study that assessed sequence variation of the universal COI barcode and variation at eight microsatellite loci to address genetic divergence in the R. miraletus species complex. The results from the study suggest several species actually occur within the R. miraletus group and the authors present a compelling argument for this case. The manuscript is clear and concise, the methods and results are presented in a straightforward and easily digestible manner, and the discussion ties everything together. In all, they did a great job, and as such, my comments are minimal and minor.
Minor Comments:
Lines 84 – 96: Raja miraletus is first mentioned in this paragraph. Here, the authors should identify the common name for the species (brown skate) because it is one of the keywords and the authors use the term brown skate in the discussion. It would be helpful for the reader to know that R. miraletus = brown skate.
Line 487: It is not clear what is meant by “definitive response”. I believe the limited sampling prevents “definitive conclusions”.
Lines 503 – 504: The term “speciation event” is used incorrectly in the context of the sentence. The authors are not assessing a speciation event, rather they are providing genetic justification for taxonomic differentiation for the two geographic populations where restricted gene flow may have led to isolation of populations and divergence, culminating in species.
Line 509: “tackled” seems an odd word choice here and is somewhat ambiguous. Can you replace this with a more direct terminology?
Table 1: It’s not clear to me what “N” (third column) is indicating. I want to assume that it implies they total number of samples, but I know that’s not a correct assumption. If “N” (third column) is the total number of samples, then it shouldn’t be shown as zero for some sites. I understand that “N (COI)” is the number of individuals sequenced at COI and “N (EST-SSRs)” is the number of individuals genotyped by microsatellite loci. So, what is “N”?
Supplemental Text S1: consider adding a reference list to the supplemental text rather than referring the reader back to the main text.
Table S5: For the scoring errors, indicate what “//” and “--" refer to. This could be done in the table caption or as a footnote.
Figure legends for the Supplemental Figurers were not included. They should be part of the manuscript package.
Author Response
We greatly thank Reviewer#1 for the positive feedback and consideration. We successfully integrated her/his hints and suggestions, as listed below.
Minor Comments:
Lines 84 – 96: Raja miraletus is first mentioned in this paragraph. Here, the authors should identify the common name for the species (brown skate) because it is one of the keywords and the authors use the term brown skate in the discussion. It would be helpful for the reader to know that R. miraletus = brown skate.
Response- We added the common name, as suggested.
Line 487: It is not clear what is meant by “definitive response”. I believe the limited sampling prevents “definitive conclusions”.
Response- We replaced “response” with “conclusions”, as suggested.
Lines 503 – 504: The term “speciation event” is used incorrectly in the context of the sentence. The authors are not assessing a speciation event, rather they are providing genetic justification for taxonomic differentiation for the two geographic populations where restricted gene flow may have led to isolation of populations and divergence, culminating in species.
Response- We replaced “speciation event” with “This diversification”, as suggested.
Line 509: “tackled” seems an odd word choice here and is somewhat ambiguous. Can you replace this with a more direct terminology?
Response- We replaced “tackled” with “influenced”.
Table 1: It’s not clear to me what “N” (third column) is indicating. I want to assume that it implies they total number of samples, but I know that’s not a correct assumption. If “N” (third column) is the total number of samples, then it shouldn’t be shown as zero for some sites. I understand that “N (COI)” is the number of individuals sequenced at COI and “N (EST-SSRs)” is the number of individuals genotyped by microsatellite loci. So, what is “N”?
Response- We replaced “N” with “N (tot)”, referring to the total number of individuals sampled in this study, according to the methods indicated in the Source column. Of these, many have been COI-sequenced [N (COI)], and genotyped [N (EST-SSRs)] ex novo within this study. N (tot) = 0 when only public COI sequences obtained by other Authors were retrieved from public databases and integrated into the mtDNA dataset, as specified in the Source column. Referenced NCBI accession numbers and Bold records used for the integrated mtDNA analyses are listed in Table S1. We improved the caption of Table 1 and modified the Table, accordingly.
Supplemental Text S1: consider adding a reference list to the supplemental text rather than referring the reader back to the main text.
Response- We agree with Reviewer#1 and now we added the reference list also in the Supplementary Text S1. We followed the instructions for authors “Citations and References in Supplementary files are permitted provided that they also appear in the main text and in the reference list”.
Table S5: For the scoring errors, indicate what “//” and “--" refer to. This could be done in the table caption or as a footnote.
Response- We thank Reviewer#1 for this comment. With “//” we indicated that no scoring errors were detected by Microchecker. We modified Table S5 and relative caption, accordingly.
Figure legends for the Supplemental Figurers were not included. They should be part of the manuscript package.
Response- Figure legends of the Supplementary Materials are already listed (please see the last 11 lines of the Supplementary Materials section).
Reviewer 2 Report
The authors present an interesting study on genetic diversity and differentiation of Raja miraletus species complex. The bright side of the manuscript is that to provide practical details on current genetic and taxonomic structure of the species complex. In this context, the study contributes understanding the evolution of species complex. There are only few minor concerns are raised. Therefore, I would like to make some suggestions to improve the quality of the paper as below:
Line 49: The importance of the results and how the results contribute to further studies can be given at the end of the abstract. In my opinion, it is always good to finish the abstract with such a sentence.
Lines 144-151: Please rephase here and explain the explain the purpose of the study clearly. This part of the paper is important since the authors should explain the purpose of the study and their hypothesis (I mean; what is the problem and what did you do to solve this problem) are given here.
Lines 530-534: Please rephase here with short separate sentences.
Lines 540: A sentence that explains the how your results contribute to the taxonomy and conservation of the species.
Author Response
Response- We greatly thank Reviewer#2 for the positive evaluation. We hope to be successful in integrating her/his suggestions, as listed below.
Therefore, I would like to make some suggestions to improve the quality of the paper as below:
Line 49: The importance of the results and how the results contribute to further studies can be given at the end of the abstract. In my opinion, it is always good to finish the abstract with such a sentence.
Response- We thank Reviewer#2 for this important suggestion. We modified the abstract to integrate her/his hints and highlight our contribution to the current knowledge on the species complex.
Lines 144-151: Please rephase here and explain the explain the purpose of the study clearly. This part of the paper is important since the authors should explain the purpose of the study and their hypothesis (I mean; what is the problem and what did you do to solve this problem) are given here.
Response- We modified the paragraph to clarify the general state of the art, our initial hypothesis and the novel results obtained through the tools chosen. The structure of the paragraph has changed to welcome Reviewer#3’s suggestions as well.
Lines 530-534: Please rephase here with short separate sentences.
Response- We rephrased in separate sentences, as suggested.
Lines 540: A sentence that explains the how your results contribute to the taxonomy and conservation of the species.
Response- We added a punctual sentence on the aspect highlighted by Reviewer#2, as suggested.
Reviewer 3 Report
1、It is noted that your manuscript needs careful editing by someone with expertise in technical English editing paying particular attention to English grammar, spelling, and sentence structure so that the goals and results of the study are clear to the reader.
2、Should the title include the region in which the species are studied?
3、The preface describes a large amount of previous researches, could you use a few sentences to summarize the shortcomings of previous research and then introduce your advance (the last paragraph of the introduction)?
4、I feel that the description of the sample collection site is vague. As shown in Figure 1, the collection area is obviously divided into two regions (represented by two red rectangles respectively). But there are three divided regions in the legend (Western Indian Ocean, the Eastern-Central North-Eastern Atlantic Ocean and Mediterranean Sea). Then Table 1 is clearly divided into two large regions (Central-Southern Africa and Northeastern Atlantic-Mediterranean Sea).
5、Should the unique data in Table 2 be marked with special tags to show that it is special?
6、Table 7 shows that the genetic distance of R. riraletus between CSA and NEAM is 7.33%. Theoretically, it should be reflected in Figure 4 that there is two separate branches belong to these two sites respectively, but this does not be reflected in Figure 4.
7、The first paragraph of the discussion is more like an introduction, is it focused on that? Could you discuss your results in conjunction with previous studies?
8、The discussion section mentioned ocean current effects, could you add ocean current model maps to visualize how ocean currents affect species differentiation?
9、In addition, this paper first proposed the use of COI and SSR molecular means to study the genetic diversity of species, but the discussion involved relatively few molecules, most of the discussion on oceanographic heterogeneities, caould you add some molecules-related discussions?
Author Response
1、It is noted that your manuscript needs careful editing by someone with expertise in technical English editing paying particular attention to English grammar, spelling, and sentence structure so that the goals and results of the study are clear to the reader.
Response- Taking advantage of our native speaking co-authors, the manuscript underwent further english revision.
2、Should the title include the region in which the species are studied?
Response- We agree that indicating the targeted geographical areas in the title would be interesting; our unanimous proposal would be “To be, or not to be: that is the Hamletic question of cryptic evolution in the Eastern Atlantic and Mediterranean Raja miraletus species complex”.
3、The preface describes a large amount of previous researches, could you use a few sentences to summarize the shortcomings of previous research and then introduce your advance (the last paragraph of the introduction)?
Response- We agree with Reviewer#3 about the hypertrophy of the description of the previous studies leading to the current knowledge on the taxonomy of the species complex. We tried to be more punctual and concise, without leaving important information aside, and at the same time allowing the reader to go through this stepwise workflow. In addition, we improved the last paragraph of the Introduction section, stressing the aims and the progress of our research in comparison with the state of the art.
4、I feel that the description of the sample collection site is vague. As shown in Figure 1, the collection area is obviously divided into two regions (represented by two red rectangles respectively). But there are three divided regions in the legend (Western Indian Ocean, the Eastern-Central North-Eastern Atlantic Ocean and Mediterranean Sea). Then Table 1 is clearly divided into two large regions (Central-Southern Africa and Northeastern Atlantic-Mediterranean Sea).
Response- We thank Reviewer#3 for the helpful comment. We corrected the Figure 1 caption as: “Sampling sites of Raja miraletus species complex collected in the Northeastern Atlantic-Mediterranean and the Central-southern Africa”.
5、Should the unique data in Table 2 be marked with special tags to show that it is special?
Response- We indicated the mean genetic distance between NEAM and CSA in bold.
6、Table 7 shows that the genetic distance of R. riraletus between CSA and NEAM is 7.33%. Theoretically, it should be reflected in Figure 4 that there is two separate branches belong to these two sites respectively, but this does not be reflected in Figure 4.
Response- Supposing that Reviewer#3 is referring to Table 2, we agree that the difference between CSA and NEAM should be reflected in Figure 4 (Bayesian coalescent tree). In our opinion, this is evident when considering the highly significant Posterior Probability (= 1) on the node corresponding to the dichotomy between haplotypes 1-23 characterising CSA, and haplotypes 24-39 characterising NEAM. We propose a modified version of Figure 4 to better highlight this differentiation. In addition, we propose a new version of Figure 2 (haplotype network), for congruence.
7、The first paragraph of the discussion is more like an introduction, is it focused on that? Could you discuss your results in conjunction with previous studies?
Response- We agree with Reviewer#3 about the redundancy of the first paragraph. We decided to delete that part, moreover considering that some of the references reported are discussed later in the same section.
8、The discussion section mentioned ocean current effects, could you add ocean current model maps to visualize how ocean currents affect species differentiation?
Response- In support of the hypothesis that ocean currents have contributed to limiting gene flow and therefore could influence species differentiation, we cited four important studies (Henriques et al. 2014, 2021; Hirschfeld et al. 2021; Last and Seret 2016). We added in the main text the explicitly reference to the Figure 2d and 2f of Hirschfeld et al. (2021) and Figure 1 of Henriques et al. (2014) which clearly explain and visualise the situation of the ocean currents and bathymetry as barriers for elasmobranchs in the Southeastern Atlantic and Mediterranean Sea. Accordingly, we modify the map in Figure 1 by adding 1) bathymetry (continental slope and shelf), 2) main oceanic currents of the East Atlantic.